# Caspase-4 dimerisation and D289 auto-processing elicit an interleukin-1β-converting enzyme

Amy H Chan[1], Sabrina S Burgener[1], Kassandra Vezyrgiannis[2], Xiaohui Wang[1], Jadie Acklam[2], Jessica B Von Pein[1], Malvina Pizzuto[1,3], Larisa I Labzin[1], Dave Boucher[2,*], Kate Schroder[1,*]

The noncanonical inflammasome is a signalling complex critical for cell defence against cytosolic Gram-negative bacteria. A key step in the human noncanonical inflammasome pathway involves unleashing the proteolytic activity of caspase-4 within this complex. Caspase-4 induces inflammatory responses by cleaving gasdermin-D (GSDMD) to initiate pyroptosis; however, the molecular mechanisms that activate caspase-4 and govern its capacity to cleave substrates remain poorly defined. Caspase-11, the murine counterpart of caspase-4, acquires protease activity within the noncanonical inflammasome by forming a dimer that self-cleaves at D285 to cleave GSDMD. These cleavage events trigger signalling via the NLRP3–ASC–caspase-1 axis, leading to downstream cleavage of the pro-IL-1β cytokine precursor. Here, we show that caspase-4 first dimerises then self-cleaves at two sites—D270 and D289—in the interdomain linker to acquire full proteolytic activity, cleave GSDMD, and induce cell death. Surprisingly, caspase-4 dimerisation and self-cleavage at D289 generate a caspase-4 p34/p9 protease species that directly cleaves pro-IL-1β, resulting in its maturation and secretion independently of the NLRP3 inflammasome in primary human myeloid and epithelial cells. Our study thus elucidates the key molecular events that underpin signalling by the caspase-4 inflammasome and identifies IL-1β as a natural substrate of caspase-4.

## Introduction

Noncanonical inflammasome activity is critical for innate immune responses to lipopolysaccharide (LPS) on cytosolic Gram-negative bacteria (Kayagaki et al, 2011, 2013; Shi et al, 2014). In mice, the noncanonical inflammasome activates the cysteine protease, caspase-11, whereas canonical inflammasomes activate caspase-1 (Ross et al, 2022). Caspase-4 and -5 are the human orthologs of

murine caspase-11 and are presumed to be activated by similar mechanisms and exert similar biological functions. In line with this, all three caspases of noncanonical inflammasomes (hereafter called caspase-4/5/11) detect and interact with LPS with the assistance of guanylate-binding proteins (Fisch et al, 2019; Kutsch et al, 2020; Santos et al, 2020; Wandel et al, 2020). LPS-activated caspase-4/5/11 subsequently cleaves the substrate gasdermin-D (GSDMD) to generate a GSDMD p30 fragment that inserts into the plasma membrane and forms pores. GSDMD pores trigger either ninjurin-1-dependent plasma membrane rupture and pyroptotic cell death (He et al, 2015; Kayagaki et al, 2015, 2021; Shi et al, 2015; Aglietti et al, 2016; Ding et al, 2016; Liu et al, 2016) or pyroptosis-associated extrusion of neutrophil extracellular traps (Chen et al, 2018). GSDMD pores also trigger ionic flux and the resultant assembly of NLRP3–ASC–caspase-1 inflammasomes, which in turn generate active caspase-1 that cleaves pro-IL-1β and pro-IL-18 into their mature forms (Rühl & Broz, 2015; Boucher et al, 2018; Monteleone et al, 2018; Chan & Schroder, 2020).

Whereas the mechanisms controlling caspase-1 and -11 activation are now defined (Boucher et al, 2018; Lee et al, 2018; Ross et al, 2018), those controlling human caspase-4/5 remain poorly understood. It is likely that dimerisation is required for the activation of caspase-4, as this underpins the activation of related initiator caspases (caspase-1, -8, -9, -11) (Boatright et al, 2003; Boucher et al, 2018; Ross et al, 2018). Caspase-1, -8 and -9 are synthesised as inactive monomeric zymogens composed of a catalytic domain (large plus small subunits of the protease) connected to an N-terminal domain (CARD or DED). The CARD domain enables recruitment to signalling platforms (Martinon et al, 2002; Pop et al, 2006) where clustering facilitates proximity-induced dimerisation of the caspase-1/8/9 enzymatic subunits to enable caspase intrinsic proteolytic activity (Renatus et al, 2001; Boatright et al, 2003; Boucher et al, 2018). Specifically, dimerisation induces autoproteolysis within either the linker that connects the protease to its N-terminal recruitment domain (e.g., the caspase-1 CARD-domain linker; CDL), or the linker that connects the two catalytic

[1]Institute for Molecular Bioscience (IMB) and IMB Centre for Inflammation and Disease Research, The University of Queensland, St Lucia, Australia    [2]Department of Biology, York Biomedical Research Institute, University of York, York, UK    [3]Structure and Function of Biological Membranes Laboratory, Université Libre de Bruxelles, Brussels, Belgium

Correspondence: K.Schroder@imb.uq.edu.au
*Dave Boucher and Kate Schroder contributed equally to this work

subunits (interdomain linker; IDL). The impact of IDL or CDL processing on protease activity and substrate repertoire varies depending on the caspase (Pop et al, 2006; Oberst et al, 2010; Boucher et al, 2018). IDL processing broadens the caspase-1 and -8 substrate repertoire, whereas CDL processing deactivates caspase-1 by ejecting dimers from the inflammasome. Dimer detachment destabilises caspase-1, causing it to dissociate into inactive monomers (Boucher et al, 2018), whereas IDL cleavage is necessary for caspase-1 to cleave cytokines. Interestingly, IDL cleavage may be dispensable for caspase-1-induced death of murine cells (Broz et al, 2010; Dick et al, 2016), suggesting that auto-processing is not always necessary for specific functions of inflammatory caspases. A recent study identified the D289 caspase-4 IDL cleavage site as necessary for LPS-induced GSDMD cleavage in HeLa cells, and concluded that a second candidate auto-processing site (D270) was dispensable for caspase-4 functions (Wang et al, 2020).

Caspase-4 is the closest orthologue of murine caspase-11 (Baker et al, 2015; Casson et al, 2015), and LPS is reported as a direct ligand of caspase-4 and -11. Indeed, binding of their CARD domains to LPS (Shi et al, 2014) or to bacteria through guanylate-binding proteins (Santos et al, 2020; Wandel et al, 2020) facilitates the multimerization of caspase-4 monomers, although how this process activates caspase-4 proteolytic function is undetermined. Homodimerisation of caspase-4 and -11 occurs through interactions within the catalytic domain (Fuentes-Prior & Salvesen, 2004). Although dimerisation of caspase-11 is alone sufficient for it to acquire proteolytic activity (Ross et al, 2018), IDL autoproteolysis is in addition triggered to produce an active p32/p10 tetrameric species (Lee et al, 2018; Ross et al, 2018). Caspase-11 IDL-processing to generate p32/p10 is required for caspase-11 to cleave GSDMD and induce pyroptosis. Caspase-11 does not appear to be able to self-cleave at the CDL to inactivate its protease function (Ross et al, 2018).

Caspase-4 cleavage fragments of ~20 and 32 kD containing the large protease subunit are reported upon noncanonical inflammasome signalling in human macrophages (Casson et al, 2015; Wang et al, 2020). However several aspects of caspase-4 cleavage are unknown, including their location (e.g., CDL versus IDL) and the responsible protease (caspase-4 itself versus another protease). Also undetermined is whether dimerisation and auto-processing affect caspase-4 activity and substrate specificity within the noncanonical inflammasome.

This study elucidates the molecular basis of caspase-4 activation during noncanonical inflammasome signalling. We demonstrate that caspase-4 needs to dimerise and auto-process at the IDL to generate fully active caspase-4 p34/p9 and p32/p9 protease species, which then cleave GSDMD to induce pyroptosis. Surprisingly, and in contrast to murine caspase-11, the p34/p9 species of active caspase-4 can also directly cleave pro-IL-1β into its mature form.

# Results

### Dimerisation and IDL cleavage activate caspase-4

Both caspase-1 and -11 require proximity-induced dimerisation to become activated (Boucher et al, 2018; Ross et al, 2018). It is possible

the same is true for caspase-4. Given that the LPS-binding CARD domain is necessary for LPS activation of caspase-4, we hypothesised that LPS binding clusters caspase-4 monomers to facilitate proximity-induced dimerisation of the caspase-4 enzymatic subunits. The caspase-4 dimer may then gain basal catalytic activity, sufficient for auto-cleavage.

To test whether dimerisation is necessary and sufficient to induce caspase-4 activity, we used the DmrB dimerisation system. This system allows precise control of caspase dimerisation (Boucher et al, 2018) that is independent of LPS interactions with the caspase-4 CARD domain. ΔCARD-caspase-4 was expressed fused to an N-terminal V5-tagged DmrB domain (Fig 1A). We anticipated that adding the dimeriser drug, AP20187, would trigger WT caspase-4 protease activity and resultant cleavage at one or more sites (Fig 1B), and that these activities would be ablated by mutation of the catalytic cysteine (C258A). The DmrB-caspase-4 constructs were expressed in human embryonic kidney (HEK) 293T cells, and caspase activity was measured by monitoring fluorescence generated by cleavage of the Ac-WEHD-AFC peptidic substrate. Indeed, AP20187-induced dimerisation of WT caspase-4 triggered Ac-WEHD-AFC cleavage and auto-processing, and this was ablated by C258A mutation of the catalytic cysteine (Fig 1C–E). Immunoblotting with an antibody that recognises the caspase-4 large subunit showed that AP20187-induced cleavage of the full-length p43 into two smaller fragments, p34 and p32 (Fig 1B and E), which V5 Western blot identified as products of IDL cleavage (Fig 1B and E). Together, these results suggest that dimerisation induces caspase-4 basal activity, IDL auto-processing, and Ac-WEHD-AFC cleavage. Caspase-4 was previously reported to form multimers (Shi et al, 2014); it is possible that this occurs through caspase-4 dimers binding to one another to form higher-order complexes. Although our investigations do not exclude the possibility that caspase-4 forms multimers in inflammasome-signalling cells, our data indicate that caspase-4 dimers are alone sufficient for protease activity.

### Caspase-4 auto-processing at D270 and D289 is required for full protease activity against a peptide substrate

To first determine sites in the IDL that are cleaved during auto-processing, and the impact of their cleavage on caspase-4 function, we identified two putative cleavage sites, D270 (WVRD↓) and D289 (LEED↓) (Fig 1B), based on sequence similarity to the preferred cleavage sequence of (W/L)EHD↓ for caspase-4 (Thornberry et al, 1997; Kang et al, 2000). Two single-point mutations at D270 and D289, and a double mutant (IDL^uncl), were created within the V5-DmrB-caspase-4 construct (Fig 1B) and tested for activity as before. Interestingly, the D270A single-point mutant retained full Ac-WEHD-AFC cleavage activity, whereas D289A mutation markedly diminished, although did not ablate, activity (Fig 1C and D). Western blots revealed that the generation of caspase-4 cleavage fragments required the catalytic cysteine and were thus products of auto-cleavage. p34 and p32 fragments corresponded to cleavage at D289 and D270, respectively, and mutating one of these residues did not ablate processing at the other (Fig 1E). Importantly, mutation of both cleavage sites (IDL^uncl) ablated caspase-4 protease activity on Ac-WEHD-AFC (Fig 1C and D) and prevented caspase-4 self-cleavage (Fig 1E). Together, these data indicate that IDL cleavage at either site

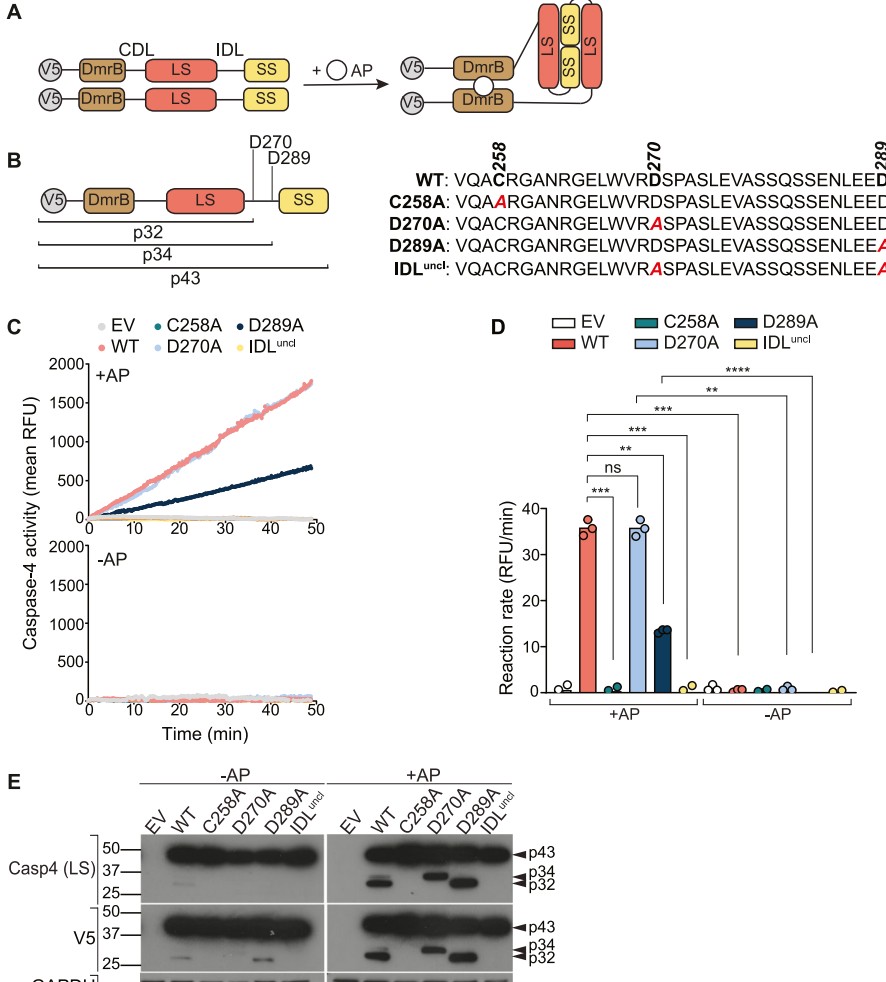

Figure 1. Dimerisation and IDL cleavage are required for caspase-4 proteolytic activity.
**(A)** Schematic of caspase-4 dimerisation with the DmrB system, which allows controlled dimerisation by AP20187 (AP). All DmrB constructs were N-terminally V5-tagged. **(B)** Schematic of band sizes generated by cleavage at D270 and D289. These putative cleavage sites were mutated (D→A) to prevent auto-processing. An IDL^uncl construct encoding mutations at both D270 and D289 sites was also included. **(C)** Caspase-4 activity was measured by relative fluorescence (RFU) generated by Ac-WEHD-AFC substrate cleavage. Caspase-4 was expressed in HEK293T cells and incubated with AP for 30 min before Ac-WEHD-AFC cleavage experiment. **(D)** Linear regression analysis (D) showing the rate of Ac-WEHD-AFC cleavage. Data are mean ± SEM of three biological replicates. Each data point represents an individual donor. $P \leq 0.01$ (**), $P \leq 0.001$ (***), $P \leq 0.0001$ (****). **(E)** Caspase-4 constructs were transfected in HEK293T cells and dimerised by AP. Auto-processing was analysed by Western blot of cell extracts.

(D289 or D270) is required for caspase-4 protease activity, with auto-processing at D289 required for full activity against the Ac-WEHD-AFC substrate. Interestingly, if cleavage at D289 is blocked by mutation, cleavage at D270 can partially compensate to generate a caspase-4 species with moderate activity. In summary, caspase-4 dimerisation induces auto-catalytic activity and self-cleavage at one or both of two IDL sites (D270, D289), and this is required for caspase-4 to cleave the Ac-WEHD-AFC substrate.

## Dimerisation and IDL auto-processing is required for caspase-4 to cleave GSDMD and IL-1β

We next sought to determine whether caspase-4 IDL auto-cleavage is required for caspase-4 proteolytic activity against its natural substrate GSDMD. Given that IDL cleavage may expose recognition sites for caspase interaction with particular substrates (Liu et al, 2020; Wang et al, 2020), we also examined whether the site of IDL cleavage affects the repertoire of substrates processed by caspase-4 by co-expressing DmrB-caspase-4 constructs alongside V5-tagged caspase substrates (GSDMD and pro-IL-1β) in HEK293T cells. AP20187 was added to induce caspase-4 dimerisation and proteolytic activity, and cleavage

of the substrates was examined. WT caspase-4 cleaved GSDMD, whereas the IDL^uncl and the catalytic mutant did not (Fig 2A). Although the D270 and D289 single mutants could cleave GSDMD, cleavage was dramatically reduced compared with WT caspase-4 (Fig 2A). This indicates that cleavage at either D270 or D289 is sufficient for GSDMD cleavage, but auto-processing at both IDL sites (D270 and D289) maximises GSDMD cleavage.

Interestingly, dimerised caspase-4 was also able to cleave pro-IL-1β to its mature 17 kD form. Pro-IL-1β was effectively cleaved by dimerised WT caspase-4 and the D270A single mutant, but not by the D289A single mutant, IDL^uncl or the catalytic mutant (Fig 2B). This suggests that caspase-4 can directly cleave pro-IL-1β if it is first dimerised and then self-cleaves at residue D289 to generate the p34/p9 active species. Taken together, these data suggest that D270 cleavage is required for caspase-4 to efficiently cleave GSDMD but not pro-IL-1β, whereas D289 auto-cleavage is critical for caspase-4 to cleave pro-IL-1β. Thus, caspase-4 self-cleavage at different IDL sites might allow for optimal recognition and processing of different substrates.

The finding that caspase-4 dimerisation induced the cleavage of human pro-IL-1β was surprising, given that caspase-11—the murine

DmrB dimerizer constructs expressed with substrates in HEK293T cells

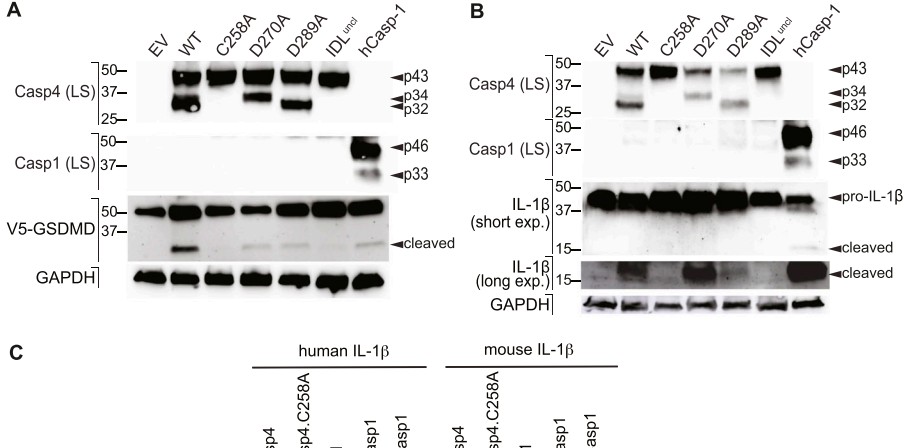

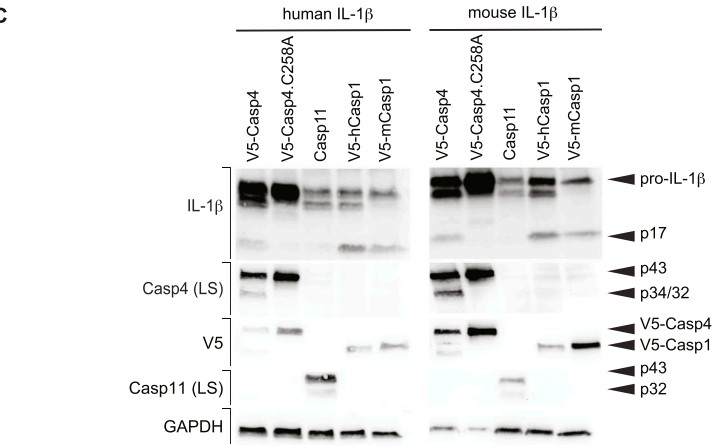

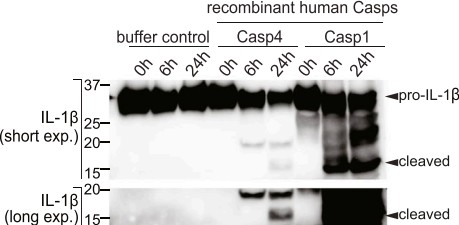

**Figure 2. Dimerisation induces caspase-4 auto-processing and cleavage of GSDMD and IL-1β in cellulo and in a fully recombinant in vitro system.**
**(A, B)** Caspase-4 constructs were co-expressed with (A) V5-GSDMD or (B) pro-IL-1β in HEK293T cells and caspase-4 was dimerised by cell exposure to AP20187. Substrate cleavage was analysed by Western blot of the whole cell lysates. **(C)** Caspase-1, 4, and 11 DmrB fusion constructs were co-expressed with human or murine pro-IL-1β in HEK293T cells, which were exposed to AP20187 to induce caspase dimerisation and activation. Substrate cleavage was analysed by Western blot of the whole cell lysates. **(D)** Purified recombinant caspase-4 or caspase-1 was incubated with purified recombinant human pro-IL-1β for 0, 6, and 24 h. **(A, B, C, D)** All data are representative of three (A, B, D) or two to four (C) biological replicates.

ortholog of caspase-4—cannot cleave murine pro-IL-1β (Bibo-Verdugo et al, 2020). We thus investigated the species specificity underpinning caspase-4-dependent pro-IL-1β cleavage by co-expressing constructs for inducible caspase dimerisation (human caspase-4, murine caspase-11, human and murine caspase-1) alongside pro-IL-1β of human or murine origin. As anticipated, dimerised caspase-1 of any species cleaved human and murine pro-IL-1β, wereas dimerised caspase-11 could not cleave pro-IL-1β of either species (Fig 2C). Dimerised caspase-4 induced the cleavage of both human and murine pro-IL-1β, and this required the caspase-4 catalytic cysteine (C258; Fig 2C). Thus, dimerisation of caspase-4, but not caspase-11, induces the cleavage of human and murine pro-IL-1β in cells.

The above overexpression studies indicate that caspase-4 activation is upstream of pro-IL-1β cleavage, but do not indicate whether caspase-4 can directly cleave pro-IL-1β. We thus employed a fully reductionist, noncellular system in which we incubated purified recombinant human caspases with purified recombinant pro-IL-1β to determine whether caspase-4 can directly cleave pro-IL-1β. All recombinant proteins (human Casp1, human Casp4, human pro-IL-1β) were sourced commercially. Recombinant caspase-4 or human caspase-1 was incubated at 37°C with recombinant pro-IL-1β for 0, 6, or 24 h. Here, caspase-4 cleaved pro-IL-1β to the bioactive 17-kD form (Fig 2D), albeit less efficiently than caspase-1, and also generated a yet undefined ~20 kD fragment. Caspase-1 also processed pro-IL-1β to a 26-kD fragment as previously reported (Afonina et al, 2015), whereas caspase-4 did not. As caspase-4 generated the mature 17 kD form of IL-1β in this fully recombinant system, this confirms caspase-4 as a bona fide IL-1β-converting enzyme.

## Cytosolic LPS induces caspase-4-dependent pro-IL-1β cleavage independently of the NLRP3 inflammasome in primary human macrophages

Transfected LPS is reported to interact directly with caspase-4 to induce caspase-4-driven death of human monocyte-derived macrophages (HMDMs), keratinocytes, and epithelial cell lines (Knodler et al, 2014; Shi et al, 2014; Casson et al, 2015). Our earlier data indicate that caspase-4 directly cleaves GSDMD (Fig 2A), which is consistent with established functions for caspase-4 (and murine caspase-11) in initiating cell death independently of caspase-1. Our data also suggested that caspase-4 directly cleaves pro-IL-1β, which is unexpected given that caspase-11 cannot. To investigate the potential for endogenous caspase-4 to directly cleave pro-IL-1β in a physiological setting, we primed HMDMs to up-regulate pro-IL-1β expression, and then transfected the cells with *Escherichia coli* (*E. coli*) K12 LPS to activate the noncanonical caspase-4 inflammasome. For comparison, nigericin was added in parallel to stimulate caspase-1 activation by the canonical NLRP3 inflammasome. Primed HMDMs were exposed to either the caspase-1/4 inhibitor VX-765 (Wannamaker et al, 2007) or the NLRP3-specific inhibitor MCC950 (Coll et al, 2019, 2022), before cell exposure to inflammasome activators. Inflammasome signalling was measured by LDH cytotoxicity assay for cell lysis, ELISA for IL-1β secretion, and Western blotting for cleaved caspase-1 and IL-1β. As expected, MCC950 suppressed nigericin-dependent cell death and IL-1β release, but not caspase-4-driven cell death induced by cytosolic LPS (Fig 3A and B). Importantly, LPS-induced secretion of mature IL-1β was minimally affected by MCC950, despite MCC950 blocking NLRP3-driven caspase-1 cleavage (Fig 3A and C), indicating that LPS-induced pro-IL-1β cleavage and release occurred independently of NLRP3 in human macrophages. The caspase-1/4 inhibitor VX-765 suppressed LPS-induced secretion of mature IL-1β (Fig 3A and C), suggesting that the NLRP3-independent response was mediated by caspase-4. We confirmed that cytosolic LPS induces caspase-4 cleavage and this was blocked by VX765 but not MCC950 (Fig 3D).

To provide further confirmation that caspase-4 can generate secreted IL-1β independently of the NLRP3 inflammasome, we exposed HMDM to *CASP4* siRNAs versus a matched control siRNA. The *CASP4* siRNA efficiently knocked down CASP4 expression without affecting CASP1 expression in HMDM (Fig 3E). *CASP4* knockdown indeed markedly suppressed HMDM IL-1β release induced by LPS transfection, demonstrating that caspase-4 mediates LPS-induced IL-1β release (Fig 3F). Importantly, *CASP4* knockdown also markedly suppressed LPS-induced IL-1β release when HMDM were treated with MCC950 to block NLRP3 inflammasome signalling; this indicates that caspase-4 drives NLRP3-independent IL-1β release (Fig 3F). LPS-induced NLRP3-independent IL-1β release in HMDM exposed to the control siRNA was ablated by the caspase-1/4 inhibitor VX765 (Fig 3F). Interestingly, individual blood donors showed some variability in LPS-induced IL-1β responses in control siRNA-treated HMDM. Some donors (n = 3) showed a minor response to MCC950 (~10% reduction) alongside a striking effect of *CASP4* knockdown (~86% reduction), indicating that caspase-4 was the primary driver of IL-1β secretion in these donors. Alternatively, in some other donors (n = 4), MCC950 reduced the LPS-induced

IL-1β response by ~50% indicating a partial contribution of NLRP3; this residual response was then ablated by *CASP4* knockdown. Thus, caspase-4 is absolutely required for IL-1β release in human primary macrophages stimulated with cytosolic LPS; here, caspase-4 performs dual functions: (i) to induce signalling by the NLRP3 inflammasome for resultant caspase-1-driven IL-1β release, and (ii) to directly cleave IL-1β for release, independently of the NLRP3 inflammasome. The latter is a major function in all our tested macrophage donors, whereas the former is only a substantial contributor in only a subset of donors. These results collectively suggest that the noncanonical inflammasome signals via distinct mechanisms in human versus murine macrophages. Coupled with our earlier data that caspase-4 can cleave pro-IL-1β in a cell-free recombinant system (Fig 2D), these findings from NLRP3-inhibited primary macrophages further support a physiological molecular function for caspase-4 as an IL-1β-converting enzyme.

## Cytosolic LPS directly induces IL-1β secretion in epithelial cells that do not express NLRP3

Caspase-4 is strongly expressed within epithelial barriers with important defence functions, such as bronchial epithelial cells and intestinal epithelial cells. Unlike myeloid cells that express NLRP3 and can induce NLRP3 signalling downstream of caspase-4, primary human epithelial cells are generally devoid of NLRP3 expression (Fig S1). To further investigate physiological functions for caspase-4 in IL-1β processing and secretion, we examined caspase-4 signalling in a human bronchial epithelial cell line (HBEC-KT) which does not express NLRP3 (Fig 4A and B). We primed HBEC-KT cells with LPS to up-regulate pro-IL-1β expression, and then after priming, cells were exposed to caspase-1/4 (VX-765) and NLRP3 (MCC950) inhibitors. Cells were then transfected with *E. coli* K12 LPS to activate the caspase-4 inflammasome. LPS transfection-induced HBEC-KT death and IL-1β secretion regardless of the presence of MCC950 (Fig 4C and D), confirming that NLRP3-activated caspase-1 was not responsible for either of these signalling outputs. The caspase-1/4 inhibitor VX-765 suppressed LPS-induced caspase-4 cleavage (Fig 4E) and IL-1β secretion, confirming that caspase-4 triggered IL-1β production.

We further confirmed NLRP3-independent IL-1β secretion in another epithelial cell line, primary human fetal intestinal epithelial cells (HIEC-6). Like HBEC-KT, HIEC-6 cells do not express NLRP3 (Fig 4F), and so do not respond to the NLRP3 agonist, nigericin (Fig 4G and H). HIEC-6 cells were primed with IFNγ and transfected with LPS to activate caspase-4. Cytosolic LPS again induced IL-1β secretion in a manner blocked by the caspase-1/4 inhibitor VX-765 (Fig 4G and H). In keeping with the lack of NLRP3 expression in HIEC-6 cells (Fig 4F), and their lack of response to the NLRP3 agonist nigericin (Fig 4G and H), the NLRP3 inhibitor MCC950 did not affect LPS-induced IL-1β secretion (Fig 4G and H), ruling out any contribution of the NLRP3-caspase-1 signalling axis for the production of IL-1β in this setting. These results, coupled with earlier data that caspase-4 can cleave pro-IL-1β in a cell-free recombinant system (Fig 2D), suggest that caspase-4 directly induces IL-1β production in HIEC-6 and HBEC-KT cells.

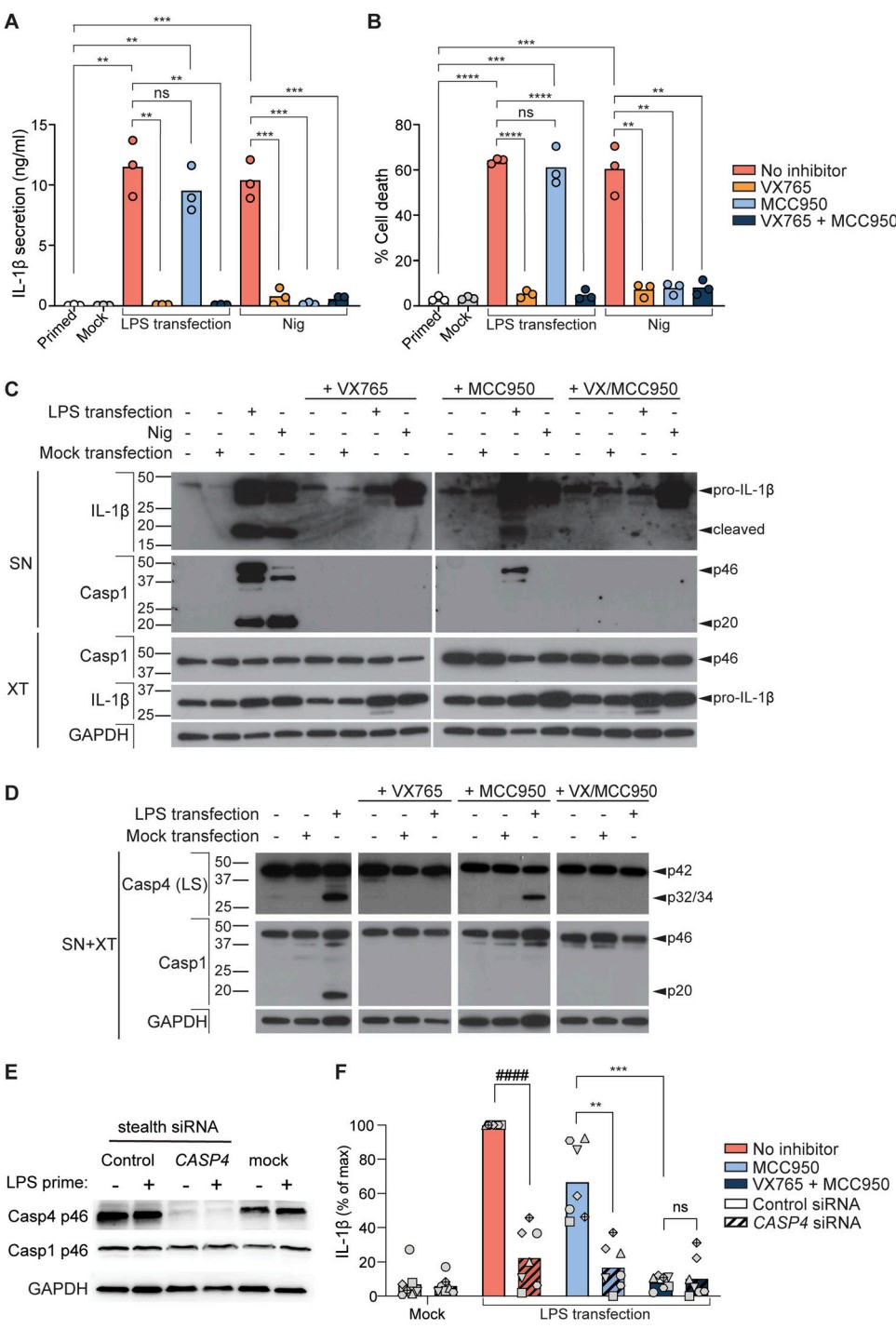

**Figure 3.  Cytosolic LPS induces CASP4-driven GSDMD and IL-1β processing independently of NLRP3 or caspase-1.**
**(A, B, C, D)** HMDMs were primed with extracellular K12 LPS (100 ng/ml) for 4 h, then either transfected with K12 LPS (10 μg/ml) using Lipofectamine LTX transfection reagent or 10 μM nigericin (Nig) was added to wells. VX-765 (10 μM) and MCC950 (10 μM) were added to cells 1 h before LPS transfection or nigericin treatment, then 4 h later, the supernatants and cell lysates were collected. **(A)** Secretion of mature IL-1β into the supernatant was assessed by ELISA (A). **(B)** Cell death was assessed by quantifying lactate dehydrogenase release into the supernatant, compared with full lysis (Triton X-100) control (B). Data (A, B) are the mean of three biological replicates with separate blood donors, and significance was assessed by unpaired $t$ test. **(C, D)** Western blots assessed pro-IL-1β cleavage in cell supernatants (SN) and pro-IL-1β expression in cell lysates (XT) (C), and caspase-1 and -4 cleavages in mixed supernatants/lysates (D). **(E, F)** HMDMs were transfected with control versus *CASP4* siRNAs and incubated for 24 h. HMDM were then primed with extracellular K12 LPS (100 ng/ml) for 4 h and then transfected with K12 LPS (10 μg/ml) using Lipofectamine LTX transfection reagent. VX-765 (10 μM) or MCC950 (10 μM) were added to cells 1 h before LPS transfection. 4 h after LPS transfection, supernatants and cell lysates were collected. **(E)** Western blots assessed the extent of CASP4 protein knockdown in LPS-primed and -unprimed cell lysates (E). **(F)** Secretion of mature IL-1β into the supernatant was assessed by ELISA (F), where data are the mean of seven experiments with separate blood donors, and significance was assessed by one-sample $t$ test (#) or paired $t$ tests (*). Each data point represents an individual donor. $P \leq 0.01$ (**), $P \leq 0.001$ (***), $P \leq 0.0001$ (****, ####). All Western blots are representative of three biological replicates with separate blood donors.

# Discussion

Innate immune responses are critical for host defence against invading pathogens, and thus require mechanisms for rapid microbial detection and response. The noncanonical inflammasome is a key signalling complex that recognises and responds to cytosolic Gram-negative bacteria. In humans, caspase-4 is constitutively expressed by many cells wherein it surveys the cytosol for bacterial LPS to sense Gram-negative bacterial infection (Knodler et al, 2014; Schmid-Burgk et al, 2015). In this study, we sought to determine the activation mechanisms of caspase-4 during non-canonical inflammasome signalling.

Previous studies suggested that LPS binding to the caspase-4 CARD domain results in caspase-4 clustering and multimerisation (Shi et al, 2014). By using the DmrB system to precisely control caspase dimerisation independently of CARD-mediated multimerisation, we

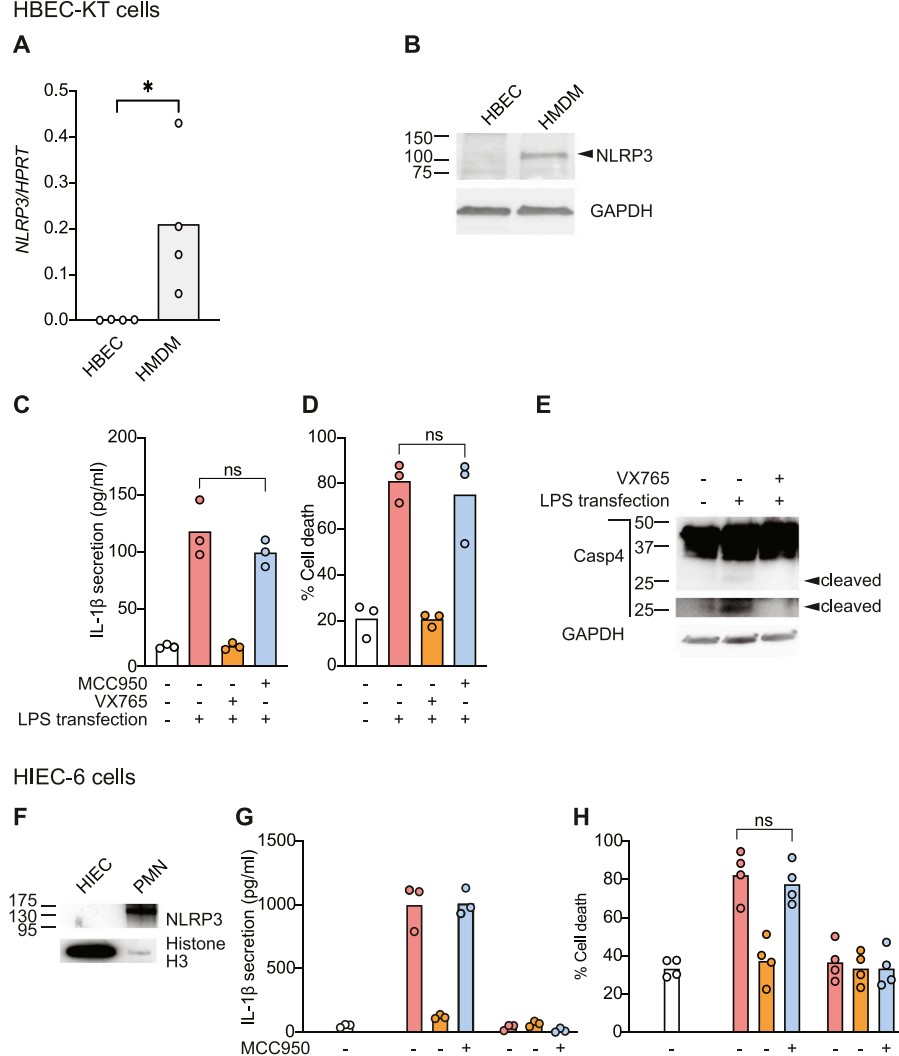

HBEC-KT cells

HIEC-6 cells

**Figure 4. Cytosolic LPS induces cell death and NLRP3-independent IL-1β secretion in human epithelial cells.**
**(A, B)** Unstimulated HBEC-KT cells and HMDMs were compared for NLRP3 expression using (A) RT–qPCR using primers targeting exons encoding the PYD and NACHT domains to capture all NLRP3 transcripts; and (B) Western blot on cell lysates. **(C, D, E)** HBEC-KT cells were primed with extracellular K12 LPS (100 ng/ml) for 4 h and then transfected with K12 LPS (10 μg/ml) using Lipofectamine LTX transfection reagent. VX-765 (10 μM) or MCC950 (10 μM) was added to cells 1 h before LPS transfection. **(F, G, H)** HIEC-6 cells were primed with IFNγ (10 ng/ml) for 16 h, and then either transfected with K12 LPS (10 μg/ml) using Lipofectamine LTX transfection reagent or 10 μM nigericin (Nig) was added to wells. VX-765 (10 μM) or MCC950 (10 μM) was added to cells 1 h before LPS transfection or nigericin treatment. Cell supernatants and cell lysates were collected 4 (HMDM and HBEC-KT) or 6 h (HIEC-6) later. **(C, G)** Secretion of mature IL-1β into the supernatant was assessed by ELISA (C, G). **(D, H)** Cell death was assessed by quantifying lactate dehydrogenase release into the supernatant, compared with full lysis (Triton X-100) control (D, H). **(E)** Lysates from stimulated HBEC-KT were analysed by Western blot and are representative of three biological replicates (E). **(F)** Unstimulated HIEC-6 cells were assessed for NLRP3 protein expression by Western blot, relative to a positive control (primary human neutrophils). Data in (A, C, D, G, H) are the mean of three to four biological replicates, and significance was assessed by unpaired t test. P ≤ 0.01 (**), P ≤ 0.001 (***), P ≤ 0.0001 (****). Data in (G) are mean and technical triplicate, and are representative of three biological replicates.

showed that dimerisation alone is sufficient to confer basal activity; higher-order complexes such as oligomers are thus dispensable. This is in support of the long-standing proximity-induced dimerisation model for initiator caspases and suggests that dimerisation of the catalytic subunits is the likely initial activation step for caspase-4. We further propose that caspase-4 CARD interactions with LPS bring caspase-4 monomer catalytic subunits into close proximity within the noncanonical inflammasome, thus enabling caspase-4 dimerisation and activation that is sufficient for caspase-4 IDL auto-cleavage.

We identified two cleavage sites within the IDL that modulate caspase-4 activity. Using the DmrB dimerisation system, we demonstrated that the IDL is cleaved at D270 and D289 to produce p32 and p34 fragments, respectively. Another report also identified these IDL cleavage sites (Wang et al, 2020), reporting that D289 cleavage is necessary for LPS-induced caspase-4-driven GSDMD processing in HeLa cells and in recombinant cleavage assays, whereas D270 was dispensable. Our results differ slightly from that

of Wang and colleagues, as we find that preventing D270 cleavage modestly decreases the capacity of caspase-4 to cleave GSDMD. Indeed, the cleavage sites D270 and D289 are flanked by two important residues—W267 and V291—that are important for hydrophobic interaction with GSDMD (Wang et al, 2020). Cleavage at both D270 and D289 is thus likely to unmask the exosite for efficient GSDMD processing. Importantly, the former study by Wang and co-workers did not investigate alternative caspase-4 substrates such as pro-IL-1β.

Our investigations reveal that cleavage at D270 and D289 bestow different degrees of caspase-4 activity and substrate selectivity, and identify pro-IL-1β as a novel substrate of caspase-4. Earlier studies suggested that recombinant caspase-4 may cleave mouse IL-1β in vitro (Kamens et al, 1995; Devant et al, 2021), but until now, caspase-4 is generally considered to be unable to cleave human pro-IL-1β in physiologically relevant systems (Bibo-Verdugo et al, 2020; Downs et al, 2020; Devant et al, 2021). Here, we demonstrate

that caspase-4 directly cleaves human IL-1β in both recombinant and cellular systems. Using the DmrB dimerisation system, we showed that IDL D289 cleavage is required for pro-IL-1β processing in cells, whereas D270 appears to be dispensable, suggesting that pro-IL-1β is cleaved by the caspase-4 species p34/9, and perhaps p32/p9. Furthermore, purified recombinant caspase-4 cleaved purified recombinant human pro-IL-1β. Using human primary macrophages and two different human epithelial cell lines, we further show that caspase-4 is responsible for NLRP3-independent IL-β production in fully endogenous cellular systems. NLRP3 expression is generally restricted to the myeloid compartment (Guarda et al, 2011), whereas caspase-4 is expressed more broadly, including at barrier surfaces that regularly encounter bacteria. The capacity for caspase-4 to initiate both pyroptosis and IL-1β maturation could be particularly important in non-myeloid cells, such as epithelial cells, that are unable to signal via NLRP3. The IL-1β-converting enzyme activity of caspase-4 thus allows these cells to mount strong and rapid pro-inflammatory responses at a site of barrier compromise, thereby preventing bacterial dissemination.

This study found that the activation mechanisms of human caspase-4 have several parallels and differences to the elucidated mechanisms of murine caspase-11. For both caspases, dimerisation of catalytic subunits appears to be the initial step for achieving proteolytic activity. Caspase-4 and caspase-11 both acquire basal protease activity upon dimerisation, whereas higher-order complexes such as multimers are dispensable. Our data further suggest that the activation mechanisms and molecular functions of caspase-4 differ from caspase-11 in two significant ways. Firstly, caspase-4 facilitates NLRP3-independent IL-1β production by directly cleaving human pro-IL-1β into its mature, active form. Secondly, two auto-processing sites within the caspase-4 IDL enable the potential generation of at least four different active dimeric caspase-4 species with distinct functions: (i) full length p43, which can auto-process; (ii) D270-cleaved p32/p11, which has moderate GSDMD-cleavage activity; (iii) D289-cleaved p34/p9, which has moderate GSDMD-cleavage activity and IL-1β convertase activity; and (iv) D270/D289-cleaved p32/p9, which has strong GSDMD-cleavage activity. This differs to caspase-11, which harbours only one IDL cleavage site (Ross et al, 2018), and which cannot process IL-1β. Similarly, pro-IL18 has been found to be a substrate for caspase-4 but not caspase-11 (Santos et al, 2020; Wandel et al, 2020). As a further complexity, caspase-4 is not the only ortholog of caspase-11; caspase-5 is a second human ortholog of caspase-11, suggestive of gene duplication in the human lineage since the last common ancestor (Schroder et al, 2012). The regulation of caspase-4 and -11 is distinct within primary macrophages, with caspase-11 and -5 strongly up-regulated by extracellular LPS, whereas caspase-4 is basally expressed and not further regulated by extracellular LPS (Schroder et al, 2012). Given that pathogens evolve quickly, the immune system is under strong selection pressure to co-evolve; indeed, immune genes are amongst the fastest to evolve within individual genomes (Mestas & Hughes, 2004). In all, these key differences between murine caspase-11 and human caspase-4 and -5 suggest that the functions of pro-inflammatory caspases have diverged across evolution, and indicate that although caspase-4 has retained its ancestral function of activating GSDMD, it has also evolved the capacity to directly cleave pro-IL-1β. Such a

function may be particularly important at human barrier surfaces, where epithelial cells commonly lack NLRP3 inflammasome function because of undetectable NLRP3 expression.

In summary, this study presents the first detailed mechanistic model for caspase-4 activation and its function on the non-canonical inflammasome. The mechanistic insight into caspase-4 activation pathways described herein delivers new understanding of the molecular processes controlling protective responses during infection, and the pathological inflammatory responses driven by noncanonical inflammasome signalling.

# Materials and Methods

### Cell lines

All cells were cultured in humidified incubators at 37°C and with 5% $CO_2$. HEK293T (CRL-3216; ATCC) cells were cultured in DMEM (Gibco) supplemented with 10% heat-inactivated FBS and 1% penicillin–streptomycin (Pen-Strep). HEK293T cells were used to express the DmrB constructs and were seeded in 10-cm cell culture dishes. The DrmB constructs were cloned into pEF6 vectors (Invitrogen) and then transfected into HEK293T cells using lipofectamine 2000 (Thermo Fisher Scientific) transfection reagent. For in cellulo substrate cleavage experiments, substrate-containing plasmids (V5-GSDMD, IL-1β) were co-transfected into HEK293T cells alongside caspase vectors. After overnight incubation at 37°C, the cells were re-seeded at a density of $1 \times 10^6$ cells/ml. HBEC-KT (CRL-4051; ATCC) cells were cultured in Airway Epithelial Cell Basal Medium (ATCC) supplemented with Bronchial Epithelial Cell Growth Kit (ATCC). HBEC-KT cells were seeded at a density of $1 \times 10^6$ cells/ml and primed for 4 h with extracellular ultrapure *E. coli* K12 LPS (100 ng/ml) before inflammasome activation. HIEC-6 (CRL-3266; ATCC) cells were cultured in reduced serum media (Opti-MEM; Gobco) supplemented with 4% FBS, 1% Pen-Strep, 1xGlutaMAX, 10 ng/ml of EGF, and 20 mM HEPES (Gibco). HIEC-6 cells were primed for 16 h with IFN-γ (10 ng/ml) before inflammasome activation.

### Primary cell culture

Studies using primary human cells were approved by the UQ Human Medical Research Ethics Committee. The Australian Red Cross Blood Service provided buffy coats from anonymous, informed, and consenting adults for this research study. Human peripheral blood mononuclear cells were isolated from screened buffy coats by density centrifugation with Ficoll-Plaque Plus (GE Healthcare) followed by magnetic-assisted cell sorting, according to standard protocols (Schroder et al, 2012). Monocyte-derived macrophages were differentiated with recombinant human CSF-1 (150 ng/ml; endotoxin-free, produced in insect cells by the UQ Protein Expression Facility) at 37°C with 5% $CO_2$. On day 6 of differentiation, HMDMs were plated at a density of 800,000 cells/ml in Roswell Park Memorial Institute (RPMI; Gibco) 1640 media supplemented with 10% heat-inactivated FBS, 1% penicillin–streptomycin, 1x GlutaMAX, and 150 ng/ml CSF-1. HMDMs were used for experimentation on day 7 of their differentiation. HMDMs were primed for 4 h with

extracellular ultrapure *E. coli* K12 LPS (100 ng/ml) before inflammasome activation.

### DmrB caspase-4 dimerisation and substrate co-expression cleavage assays

The medium from transfected HEK293T cells was replaced with Opti-MEM containing 1 $\mu$M of B/B Homodimeriser (AP20187; Clontech) and incubated for 30 min. The medium was replaced with caspase activity buffer (200 mM NaCl, 50 mM HEPES, pH 8.0, 50 mM KCl) supplemented with 100 $\mu$g/ml digitonin, 10 mM DTT, and 100 $\mu$M Ac-WEHD-AFC. Ac-WEHD-AFC is a fluorogenic substrate for caspase-1, -4, -5, and -14. Processing of Ac-WEHD-AFC was monitored at 37°C for regular time intervals using the M1000 TECAN spectrofluorometer (400 nm excitation, 505 nm emission). Cell extracts and supernatants were precipitated using methanol/chloroform and analysed by Western blot following previously described methods (Groß, 2012) and the following reagents: antibodies against the caspase-4 large subunit (mouse monoclonal, 1:1,000; Proteintech), hIL-1$\beta$ and mIL-1$\beta$ (goat polyclonal, 1:1,000; R&D Systems), caspase-1 large subunit (Bally-1, mouse monoclonal, 1:1,000; Adipogen), V5 (SV5-Pk1, mouse monoclonal, 1:2000; Abcam), and GAPDH.

### In vitro recombinant protein cleavage assays

5 U of full-length human caspase-1 (Abcam) or caspase-4 (Abcam) recombinant proteins were incubated with 2 $\mu$g of recombinant human pro-IL-1$\beta$ (Sino Biological Inc.) in caspase activity buffer (200 mM NaCl, 50 mM HEPES, pH 8.0, 50 mM KCl). The protein mix was incubated at 37°C for 0, 6, and 24 h. Cleavage of recombinant proteins was analysed by Western blot using standard methods (Groß, 2012) and the following reagents: antibodies against the caspase-4 large subunit (4B9, mouse monoclonal antibody, 1:1,000; Santa Cruz Biotechnology), human IL-1$\beta$ (goat polyclonal antibody, 1:1,000; R&D Systems), caspase-1 (D7F10, rabbit monoclonal, 1:1,000; Cell Signalling Technology).

### Caspase-4 knockdown

HMDM were seeded in a 96-well plate and incubated at 37°C with 5% $CO_2$ overnight. A stealth siRNA transfection mix was prepared as following: 500 nM stealth siRNA, 0.4% PLUS reagent (Thermo Fisher Scientific) incubated for 5 min mixed with 1% lipofectamine LTX (Thermo Fisher Scientific) and incubated for 20 min at RT. The siRNA-transfection mix was added to the cells and incubated for 24 h. siRNA efficiency was determined in unprimed and LPS-primed HMDM by immunoblot. Stealth siRNA for caspase-4 (HSS141457) and a matched stealth RNAi negative control (medium GC content: 47% GC content) were used in parallel for siRNA studies.

### Inflammasome assays

To activate the noncanonical inflammasome, cells were transfected with ultrapure *E. coli* K12 LPS (10 $\mu$g/ml) using 0.25% Lipofectamine LTX (Thermo Fisher Scientific) and following the manufacturer's instructions. To activate the NLRP3 canonical inflammasome, 10 $\mu$M nigericin sodium salt (Sigma-Aldrich) was added for 2 or 4 h. Where cells were exposed to 10 $\mu$M MCC950 or 10 $\mu$M VX-765 (MedChemExpress), these were added 1 h before the inflammasome agonists by replacing the medium with Opti-MEM replete with inhibitors. IL-1$\beta$ secretion into the supernatant was assessed by ELISA (R&D Systems), according to the manufacturer's instructions. Cell cytotoxicity was measured using the CytoTox96 Non-radioactive Cytotoxicity Assay (Promega) and expressed as a percentage of total cellular LDH (100% lysis control). Cell extracts and methanol/chloroform-precipitated supernatants were analysed by Western blot using standard methods (Groß, 2012) and the following reagents: antibodies against the caspase-4 large subunit (4B9, mouse monoclonal antibody, 1:1,000; Santa Cruz Biotechnology; or mouse monoclonal, 1:1,000; Proteintech), human IL-1$\beta$ (goat polyclonal antibody, 1:1,000; R&D Systems), caspase-1 (D7F10, rabbit monoclonal, 1:1,000; Cell Signalling Technology), GSDMD (rabbit polyclonal antibody, 1:1,000; Cusabio), NLRP3 (D4D8T Cell Signaling Technologies or ProteinTech rabbit polyclonal antibody 19771-1-AP; both 1:1,000), and GAPDH (polyclonal rabbit antibody, 1:5,000; BioScientific).

### mRNA expression analysis

Cell monolayers were harvested in 350 $\mu$l Buffer RLT with $\beta$-mercaptoethanol, and RNA was processed using the RNeasy Mini Kit (QIAGEN) with on-column DNA digestion following the manufacturer's protocol. RNA concentrations were determined on NanoDrop spectrophotometer and an equal amount of RNA from each sample was used for downstream cDNA synthesis. cDNAs were synthesised by reverse transcription using Superscript III Reverse Transcriptase (Thermo Fisher Scientific) with OligoDT priming. Gene expression was quantified in 384 well plates by RT-qPCR using SYBR Green reagent (Applied Biosystems) on a QuantStudio 7 Flex Real-Time PCR System (Thermo Fisher Scientific). Relative gene expression was determined using the change-in-threshold ($2^{-DDCT}$) method with hypoxanthine phosphoribosyltransferase 1 (HPRT) as an endogenous control.

### Statistical analysis

Statistical analysis was performed using GraphPad Prism 8.0 software. Biological replicates were pooled by combining the means of technical replicates. Data were analysed for normality using the Shapiro–Wilk normality test. For caspase activity assays, Ac-WEHD-AFC activity curves were analysed by linear regression to determine the slope (relative fluorescence units/min) then tested for statistical significance using parametric paired $t$ tests (two-sided). Data were considered significant when $P \leq 0.05$ (*), $P \leq 0.005$ (**), $P \leq 0.0005$ (***), and $P \leq 0.0001$ (****). Data from ELISA and LDH assays were tested for statistical significance using parametric unpaired $t$ tests (two-sided). Data were considered significant when $P \leq 0.01$ (*), $P \leq 0.001$ (**), $P \leq 0.0001$ (****).

## Supplementary Information

# Acknowledgements

We gratefully acknowledge Dr. Fiona Wylie for editing this article. This work was supported by the Australian Research Council (Discovery Project DP190102285), the National Health and Medical Research Council of Australia (Fellowships 1141131 and 2009075 to K Schroder; Synergy Grant 2009677 to K Schroder), The University of Queensland (Postdoctoral Fellowship to D Boucher), a Novartis Foundation for Medical-Biological Research Fellowship (#21C133 to SS Burgener), the National Fund for Scientific Research of Belgium (FNRS CR Postdoctoral Fellowship to M Pizzuto), the University of York (Generation Research Studentship to J Acklam) and a Springboard Award from the Academy of Medical Science and the Wellcome Trust (SBF006/1025 to D Boucher).

## Author Contributions

AH Chan: conceptualization, data curation, formal analysis, validation, investigation, methodology, project administration, and writing—original draft, review, and editing.

SS Burgener: formal analysis, investigation, methodology, and writing—review and editing.

K Vezyrgiannis: investigation.

X Wang: formal analysis and investigation.

J Acklam: investigation.

JB Von Pein: investigation.

M Pizzuto: formal analysis, investigation, and methodology.

LI Labzin: resources, supervision, and writing—review and editing.

D Boucher: conceptualization, resources, supervision, investigation, methodology, and writing—review and editing.

K Schroder: conceptualization, resources, data curation, formal analysis, supervision, funding acquisition, methodology, project administration, and writing—review and editing.

## Conflict of Interest Statement

K Schroder is a co-inventor on patent applications for NLRP3 inhibitors licensed to Inflazome Ltd., a company headquartered in Dublin, Ireland. Inflazome is developing drugs that target the NLRP3 inflammasome to address unmet clinical needs in inflammatory disease. K Schroder served on the Scientific Advisory Board of Inflazome in 2016–2017, and serves as a consultant to Quench Bio, USA and Novartis, Switzerland. The authors have no additional financial interests.

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
