## [Reviewer comments · Life Science Alliance]

Life Science Alliance

Caspase-4 dimerisation and D289 auto-processing elicits an interleukin-1 β converting enzyme

Amy Chan, Sabrina Burgener, Kassandra Vezyrgiannis, Xiaohui Wang, Jadie Acklam, Jessica von Pein, Malvina Pizzuto, Larisa Labzin, Dave Boucher, and Kate Schroder

DOI: <https://doi.org/10.26508/lsa.202301908>

Corresponding author(s): Kate Schroder, University of Queensland

Review Timeline:

Submission Date:	2023-01-09
Editorial Decision:	2023-02-28
Revision Received:	2023-07-08
Editorial Decision:	2023-07-24
Revision Received:	2023-07-24
Accepted:	2023-07-27

Scientific Editor: Novella Guidi

Transaction Report:

February 28, 2023

Re: Life Science Alliance manuscript #LSA-2023-01908

Prof. Kate Schroder
University of Queensland
Institute for Molecular Bioscience
St Lucia
The University of QLD
Brisbane 4072
Australia

Dear Dr. Schroder,

Thank you for submitting your manuscript entitled "Caspase-4 dimerisation and D289 auto-processing elicits an interleukin-1 β converting enzyme" to Life Science Alliance. The manuscript was assessed by expert reviewers, whose comments are appended to this letter. We invite you to submit a revised manuscript addressing the Reviewer comments.

Thank you for this interesting contribution to Life Science Alliance. We are looking forward to receiving your revised manuscript.

Sincerely,

B. MANUSCRIPT ORGANIZATION AND FORMATTING:

Reviewer #1 (Comments to the Authors (Required)):

In this study, Chan and colleagues present convincing data on the molecular basis of how caspase-4 undergoes dimerization and auto-processing to become fully active. Although the multimerization of caspase-4 has been previously reported, the authors make the important observation that dimerization is necessary and sufficient for caspase-4 activation. They identified the cysteines D270 and D289 in the interdomain linker as essential residues required for caspase-4 auto-processing. They also reported that caspase-4 requires auto-processing at both D270 and D289 cysteines for the cleavage of GSDMD. In contrast, D289 cleavage is necessary for pro-IL-1 β processing, while D270 processing is dispensable. The observation that caspase-4 can directly process pro-IL-1 β is striking since the mouse orthologue caspase-11 cannot do so. Finally, it was demonstrated that, upon LPS transfection, caspase-4-mediated processing of pro-IL-1 β occurs independently of NLRP3 in human epithelial cell lines. Overall, the data are interesting and fully support the conclusions.

The following minor points need to be improved.

- The authors should discuss whether there might be the possibility that the caspase-4 dimers bind each other to form multimers.
- The Ac-WEHD-AFC peptide substrate used to assess caspase-4 activity is a fluorogenic substrate specific not only for caspase-4 but also caspase-1, -5, and -14. The authors should clarify this in the method description.
- Fig. 1C and D: the color pattern of the WT and C258A conditions is very similar, making these conditions indistinguishable.

Reviewer #2 (Comments to the Authors (Required)):

Summary

The current study by Chan, et al. sets out to understand how human Caspase-4 (ortholog of murine Caspase-11) is activated and how such activation leads to processing of interleukin 1 β . The authors conclude that Caspase-4 directly cleaves IL-1 β independent of the inflammasome (NLRP3). In contrast to their previous study using a murine model with Caspase-11 that IL-1 β activation was NLRP3-dependent, here they conclude that IL-1 β activation by Caspase-4 is independent of the inflammasome. The authors have an enviable track-record at solving inflammasome mechanisms. However, the current study leaves one not quite satisfied with the evidence for a distinct mechanism. If the authors could provide additional evidence for this independence, the manuscript would be of more interest to the community.

Major Issues

Issue 1: The same group of authors has previously identified the homologous auto-processing activation of caspase-11 in mice (the caspase-4 ortholog). The authors' 2018 LSA paper entitled "Dimerization and auto-processing induce caspase-11 protease activation within the non-canonical inflammasome" proved out everything in the present study except for the key difference in concluding that caspase cleavage of IL-1 β requires the inflammasome in mice but not in human cells. Given the similarities across mammalian caspase-activation pathways, this finding, though intriguing, is not particularly convincing. The title suggests that caspase-4 is an ICE enzyme. The authors do not have compelling evidence that Caspase-4 is acting directly on IL-1 β . As such, the current study comes across as identifying something novel but is rather vague as to what that might be.

Issue 2: The evidence for IL-1 β activation independent of NLRP3 is not convincing. The authors provide evidence that their cell line of choice does not express NLRP3 (Fig 4A). Using one primer set and qRT-PCR to say the HBECs have no detectable NLRP3 is not evidence for NLRP3 independence. This is particularly concerning for human NLRP3 that has well-established isoform switching. Were all isoforms assessed? Even if expressed at undetectable levels, this assay does not provide strong evidence for independence. A null allele of NLRP3 would provide stronger evidence. What about other cell lines? Is this peculiar to HBECs?

Issue 3: Throughout the manuscript, the authors conclusions are not directly supported by the evidence provided and some leaps in logic are implied. Revising the writing would help the reader.

Example 1, Abstract: "although the molecular mechanisms that activate caspase-4 and govern its capacity to cleave substrates are poorly defined". The authors' current and previous papers address the activation of caspase but not regulation of substrate interaction or substrate selection. Therefore, "the capacity to cleave substrates" has not been addressed.

Example 2, Abstract: "acquires protease activity within the non-canonical inflammasome by forming a dimer that self-cleaves at D285 to directly cleave GSDMD". Auto-activation of caspase by cleavage at D285 does not inform us that subsequent cleavage of GSDMD is "direct". The evidence provided only supports that auto-cleavage of the caspase at D285 is necessary for GSDMD cleavage.

Reviewer #3 (Comments to the Authors (Required)):

In This manuscript, Chan et al, by using the DmrB dimerization system, found that caspase-4 dimerizes then self-cleaves at two sites - D270 and D289 - in the interdomain linker(IDL) to drive GSDMD-dependent cell lysis. Interestingly, the author also reports that self-cleavage at D289 generates a caspase-4 p34 form that directly cleaves pro-IL-1b. This is in contrast to its murine orthologue, caspase-11, which doesn't directly cleave pro-IL-1 β into its mature form, but relies on the associated NLRP3-activation-to drive caspase-1 dependent cleave of pro-IL-1b. Using HMDMs and HBEC-KT cells, the author further found that Cytosolic LPS induces GSDMD and IL-1 β processing independently of NLRP3 or caspase-1. Overall, this is a good-quality study; the experiments are well performed and the data are strong, providing a new perspective to understanding the mechanism of activation of caspase-4. However, there are still a few gaps and limitations that need to be addressed in order to firmly establish the mechanism proposed by the authors.

The authors set up a robust in vitro assay to examine the activation of CASP4. We can clearly see that the Cytosolic LPS induces GSDMD-pyroptosis and pro-IL-1 β cleavage independently of the NLRP3 inflammasome. However, the current data have not firmly established direct evidence that Caspase-4 dimerizes and directly cleaves pro-IL-1b is lacking:

- In fig 3 can the author examine the oligomerization/ dimerization of caspase-4 using the native gel? Can the author also use shRNA or siRNA to support the claim that caspase-4 activation in macrophages directly cleaves IL-1b?
- In Fig. 3C active Caspase 4 blot is needed to show the cleavage of caspase-4 in macrophages.
- In Fig.4D it would be very informative if the author would include caspase1 and IL-1b in the western blot.
- In Fig.1C. the recombinant caspase-4 cleaves IL-1b with slow kinetic? Why is that? Also, is it possible to examine this using 293T cell overexpression of full-Caspase 4 and IL-1b

A previous study (Ref Wang et al., 2020) has already shown that D289 cleavage is necessary for LPS-induced caspase-4 signaling and GSDMD processing, whereas D270 is partially dispensable. The authors mentioned this in the discussion but should credit more the previous study and explains it in more detail in the introduction session.

We thank the reviewers for their thorough and insightful reviews of our manuscript, for their valuable suggestions and their positive comments. In this revised manuscript we have addressed points raised by the reviewers and we now include seven new data panels substantiating, and extending on, our original findings. Major changes to our study, as suggested by Reviewers, are highlighted in yellow in the manuscript documents. We feel that in addressing the points raised in peer review we have significantly enhanced the impact and clarity of our manuscript, and provided further mechanistic insight into CASP4 activation and functions. Please find below a point-by-point reply to each comment.

Reviewer #1:

In this study, Chan and colleagues present convincing data on the molecular basis of how caspase-4 undergoes dimerization and auto-processing to become fully active. Although the multimerization of caspase-4 has been previously reported, the authors make the important observation that dimerization is necessary and sufficient for caspase-4 activation. They identified the cysteines D270 and D289 in the interdomain linker as essential residues required for caspase-4 auto-processing. They also reported that caspase-4 requires auto-processing at both D270 and D289 cysteines for the cleavage of GSDMD. In contrast, D289 cleavage is necessary for pro-IL-1b processing, while D270 processing is dispensable. The observation that caspase-4 can directly process pro-IL-1b is striking since the mouse orthologue caspase-11 cannot do so. Finally, it was demonstrated that, upon LPS transfection, caspase-4-mediated processing of pro-IL-1b occurs independently of NLRP3 in human epithelial cell lines. Overall, the data are interesting and fully support the conclusions.

We thank the Reviewer for their positive comments and support for our study, which investigates how dimerization and self-cleavage at specific aspartic acid residues (D270, D289) modulates caspase-4 molecular functions.

The following minor points need to be improved.

- The authors should discuss whether there might be the possibility that the caspase-4 dimers bind each other to form multimers.

We have incorporated this excellent suggestion in line 112 of our revised document, which now reads *“Caspase-4 was previously reported to form multimers (Shi et al., 2014); it is possible that this occurs through caspase-4 dimers binding to one another to form higher-order complexes. While our investigations do not exclude the possibility that caspase-4 forms multimers in inflammasome-signalling cells, our data indicate that caspase-4 dimers are alone sufficient for protease activity.”*

- The Ac-WEHD-AFC peptide substrate used to assess caspase-4 activity is a fluorogenic substrate specific not only for caspase-4 but also caspase-1, -5, and -14. The authors should clarify this in the method description.

We have incorporated this excellent suggestion in line 355 of our revised methods section, which now includes the sentence: *“Ac-WEHD-AFC is a fluorogenic substrate for caspase-1, -4, -5 and -14.”*

- Fig. 1C and D: the color pattern of the WT and C258A conditions is very similar, making these conditions indistinguishable.

We thank the Reviewer for bringing this to our attention. We have now changed the colour of C258A to green in Fig 1C-D for greater clarity.

Reviewer #2:

The current study by Chan, et al. sets out to understand how human Caspase-4 (ortholog of murine Caspase-11) is activated and how such activation leads to processing of interleukin 1beta. The authors conclude that Caspase-4 directly cleaves IL-1beta independent of the inflammasome (NLRP3). In contrast to their previous study using a murine model with Caspase-11 that IL-1beta activation was NLRP3-dependent, here they conclude that IL-1beta activation by Caspase-4 is independent of the inflammasome. The authors have an enviable track-record at solving inflammasome mechanisms. However, the current study leaves one not quite satisfied with the evidence for a distinct mechanism. If the authors could provide additional evidence for this independence, the manuscript would be of more interest to the community.

We thank the Reviewer for their positive comments and support for our study, which shows that unlike caspase-11 in mice, caspase-4 in humans directly cleaves IL-1 β independently of the NLRP3 inflammasome. The revised manuscript provides substantial further evidence supporting NLRP3-independent CASP4-driven IL-1beta cleavage in cells, as well as evidence of direct cleavage of IL-1beta by CASP4 in a cell-free, fully recombinant system, as detailed below.

Major Issues

Issue 1: The same group of authors has previously identified the homologous auto-processing activation of caspase-11 in mice (the caspase-4 ortholog). The authors' 2018 LSA paper entitled "Dimerization and auto-processing induce caspase-11 protease activation within the non-canonical inflammasome" proved out everything in the present study except for the key difference in concluding that caspase cleavage of IL-1beta requires the inflammasome in mice but not in human cells.

Indeed, we previously elucidated caspase-11 activation mechanisms in our LSA article (Ross *et al.*, 2018). Data from this earlier study is consistent with an activation mechanism of the following steps: (1) LPS induces the dimerisation of caspase-11, (2) caspase-11 dimerisation elicits basal protease activity, such that caspase-11 can self-cleave at a single site within the IDL (D285) to generate a fully active caspase-11 species (p32/p10) that cleaves gasdermin D to induce macrophage death and NLRP3-dependent IL-1 β production. Thus, there are only two species of active caspase-11 dimers (a full length dimer that can only cleave itself, and the p32/p10 dimer that can cleave GSDMD) and a single natural substrate for this protease (GSDMD).

The current study elucidating caspase-4 activation mechanisms suggests an activation model with key similarities and differences to that of caspase-11. As for caspase-11, we propose that

LPS induces caspase-4 dimerisation, which allows caspase-4 to acquire the basal protease activity required for self-cleavage. Unlike caspase-11 however, caspase-4 can self-cleave at one or both of two IDL sites – D270 and D289. Here, the existence of two IDL self-cleavage sites allows caspase-4 to generate up to three additional active species with different substrate specificities: (i) D270-cleaved p32/p11, which has moderate GSDMD-cleavage activity, (ii) D289-cleaved p34/p9, which has moderate GSDMD-cleavage activity and IL-1 β convertase activity, and (iii) D270/D289-cleaved p32/p9, which has strong GSDMD-cleavage activity. **This differs to caspase-11, which harbours only one IDL cleavage site, can only generate a single fully active species, and cannot process pro-IL-1 β (Ross *et al.*, 2018). Thus caspase-4, but not caspase-11, generates several active species with distinct substrate specificities.**

Given the similarities across mammalian caspase-activation pathways, this finding, though intriguing, is not particularly convincing.

While mammalian caspase-activation pathways are broadly similar there are key differences between the activation mechanisms and functions of individual inflammatory caspases, within and between species (as reviewed by us in Ross *et al.*, 2022 *Annual Reviews Immunology*). As one published example of functional differences between murine caspase-11 and human caspase-4, pro-IL18 is cleaved by caspase-4 but not caspase-11 (Santos *et al.*, 2020; Wandel *et al.*, 2020). As a further complexity, caspase-4 is not the only ortholog of caspase-11; caspase-5 is a second human ortholog of caspase-11, suggestive of gene duplication in the human lineage since the last common ancestor (Schroder *et al.*, 2012). The gene regulation of caspase-4 and -11 is also distinct within primary macrophages, with caspase-11 and -5 expression strongly upregulated by extracellular LPS, while caspase-4 is basally expressed and not further regulated by extracellular LPS (Schroder *et al.*, 2012).

Given that pathogens evolve quickly, the immune system is under strong selection pressure to co-evolve; indeed, immune genes are amongst the fastest to evolve within individual genomes (Mestas & Hughes, 2004). In all, these key differences between murine caspase-11 and human caspase-4 and -5 suggest that the functions of pro-inflammatory caspases have diverged across evolution, and indicate that while caspase-4 has retained its ancestral function of activating GSDMD, it has also evolved the capacity to directly cleave pro-IL-1 β . Such a function may be particularly important at human barrier surfaces, where epithelial cells commonly lack NLRP3 expression and hence inflammasome function.

We agree that these differences between caspase-4 and caspase-11 are intriguing. Our revised manuscript includes seven new data panels that we believe provides a compelling case for direct cleavage of pro-IL-1 β by caspase-4 (as detailed in the response to the next point)

The title suggests that caspase-4 is an ICE enzyme. The authors do not have compelling evidence that Caspase-4 is acting directly on IL-1beta. As such, the current study comes across as identifying something novel but is rather vague as to what that might be.

Our revised manuscript includes seven new data panels, which alongside our original data, provides a compelling case for direct cleavage of pro-IL-1 β by caspase-4. In summary, we have used an extensive suite of biochemical, cellular and synthetic biology methods to demonstrate that caspase-4 directly generates the mature, secreted form of IL-1 β independently of the NLRP3 inflammasome:

1. We engineered a DmrB-caspase-4 fusion protein to precisely control caspase-4 dimerisation, auto-processing and substrate cleavage in cells; using this system we showed that:
 - a. Dimeric caspase-4 auto-processed at the D289 residue (p34/p9 species) induces the maturation of human pro-IL-1 β (**Fig 2B**)
 - b. Caspase-4 dimers require the catalytic cysteine of their protease domain (C285) in order to induce cleavage of human and murine pro-IL-1 β (**new Fig 2C**), confirming that the protease activity of caspase-4 drives pro-IL-1 β cleavage.
 - c. In side-by-side comparisons, dimeric caspase-4 and caspase-1 induce cleavage of human and murine pro-IL-1 β , while caspase-11 cannot; confirming the known capacity of caspase-1 but not caspase-11 to cleave pro-IL-1 β (**new Fig 2C**) and revealing that caspase-4 has a distinct function compared to its murine ortholog caspase-11.
2. We used a fully recombinant, cell-free system to confirm that caspase-4 directly cleaves pro-IL-1 β (**Fig 2D**). Here, we incubated recombinant purified caspase-4 with recombinant purified pro-IL-1 β (all sourced commercially) in a tube with nothing other than buffer; in this fully recombinant system caspase-4, like caspase-1, directly cleaved pro-IL-1 β . Given that this cleavage assay is a fully reductionist system – the only proteins present are caspase-4 and its substrate pro-IL-1 β – we consider this the gold standard assay for demonstrating that caspase-4 directly cleaves this substrate. We agree that caspase-4 is not as efficient as caspase-1 to cleave pro-IL-1 β in this reductionist system, but these data show that caspase-4 is nonetheless an ICE.

We then show that direct cleavage of pro-IL-1 β by caspase-4 is physiologically relevant in the cellular systems for which caspase-4 function is known to be critical for host defence – macrophages and epithelial cells:

3. We show that caspase-4 is critically required for IL-1 β secretion (**Fig 3A, Fig 3C, new Fig 3F**) and cleavage (**Fig 3C**) in human monocyte-derived macrophages (HMDM) challenged with cytosolic LPS. HMDMs express NLRP3, and non-canonical inflammasome signalling is well appreciated to trigger a second wave of signalling by the NLRP3-ASC-caspase-1 inflammasome. Thus, these experiments focus on NLRP3-independent IL-1 β responses by pretreating HMDM with MCC950 to disable NLRP3-induced caspase-1 activation (verified in **Figs 3C, new Fig 3D**). We demonstrate the critical requirement for caspase-4 in this NLRP3-independent IL-1 β response using a pharmacological approach (the caspase-1/4 inhibitor VX765, which will only block caspase-4 signalling in MCC950-treated cells that have already silenced the NLRP3-ASC-caspase-1 inflammasome; **Fig 3A, Fig 3C, new Fig 3D, new Fig 3F**) as well as a genetic approach (specific RNA silencing of caspase-4 in MCC950-treated cells, **new Figs 3E-3F**). All approaches consistently show that cytosolic LPS triggers substantial NLRP3-independent IL-1 β release, and that caspase-4 is required for this response.
4. We next explore caspase-4 signalling in epithelial cell lines. Human epithelial cells commonly lack *NLRP3* expression (**new Fig S1** and below, data sourced from The Human Protein Atlas), so offer a useful system for studying NLRP3-independent caspase-4 responses, and one with important physiological relevance for barrier defence against Gram-negative bacteria. We employed two distinct human epithelial lines (human bronchial epithelial cells, HBEC-KT; primary human fetal intestinal epithelial cells, HIEC-6). Like other primary human epithelial cells (**Fig S1**), these two cell lines are devoid of *NLRP3* mRNA and protein expression (**new Figs 4A, 4B, 4F**). As such, we anticipated caspase-4 would signal in these cells without engagement of

the NLRP3 inflammasome pathway. Indeed, these cells do not respond to the NLRP3 inflammasome agonist nigericin (**Fig 4G-H**) and cytosolic LPS responses are unaffected by the NLRP3 inflammasome inhibitor MCC950 (**Fig 4C-E, 4G-H**). Thus, when these cells are challenged with cytosolic LPS, the ensuing release of IL-1 β is entirely independent of the NLRP3-ASC-CASP1 inflammasome. Consistent with a key function for the LPS sensor caspase-4 in driving IL-1 β release, this response was ablated by the caspase-1/4 inhibitor VX765. Thus, caspase-4 rather than NLRP3/caspase-1, is fully responsible for driving IL-1 β release in LPS-stimulated epithelial cells.

Supplementary Figure 1. Human epithelial cells from diverse sources do not express *NLRP3* mRNA, which is myeloid-specific.

Issue 2: The evidence for IL-1beta activation independent of NLRP3 is not convincing. The authors provide evidence that their cell line of choice does not express NLRP3 (Fig 4A). Using one primer set and qRT-PCR to say the HBECs have no detectable NLRP3 is not evidence for NLRP3 independence. This is particularly concerning for human NLRP3 that has well-established isoform switching. Were all isoforms assessed? Even if expressed at undetectable levels, this assay does not provide strong evidence for independence. A null allele of NLRP3 would provide stronger evidence. What about other cell lines? Is this peculiar to HBECs?

We apologise if our manuscript was not sufficiently clear on this point. We have added extra data as well as clarifying information to our manuscript to resolve the important point that our epithelial cell lines do not express NLRP3, and NLRP3 does not contribute to their LPS-induced IL-1 β response:

1. It is generally accepted in the field that human primary epithelial cells from diverse sources do not express NLRP3 transcript or protein. To evidence this important point, we now include single cell RNA seq data from the Human Protein Atlas (**Fig S1**, also above) that shows that 34 individual human primary epithelial cell types do not express *NLRP3*. Thus, HBEC are not unusual in their lack of *NLRP3* expression; this is a general feature of primary human epithelial cells (and one we also observe in HIEC-6). For all further analyses of epithelial cells we examined not one but two different epithelial cell lines of different origin (bronchial versus intestinal) to ensure the generality of our findings.
2. Our qPCR primer design targets two exons to prevent gDNA amplification. Figure 4A *NLRP3* qPCR from our original submission used primers spanning two exons present in all documented NLRP3 transcripts (see locus figure below). However, we take the Reviewer's point that NLRP3 is subject to intense splice variation, particularly within the final exons encoding the leucine-rich repeats (some of which may be dispensable

for NLRP3 function). Thus, we have reanalysed NLRP3 expression with a second set of qPCR primers designed to target the early exons that encode the pyrin and NACHT domains critical for NLRP3 signalling; these two exons are contained by all known NLRP3 spliceforms, and any yet-unidentified spliceforms lacking these domains will not encode an NLRP3 variant that can assemble an inflammasome. qPCR profiles for HBEC-KT epithelial cells versus human primary macrophages (HMDM) are near-identical in our repeat assay, with both primer sets showing that HMDM express substantial NLRP3 mRNA while HBEC do not (see below). We have replaced Fig 4A in the revised manuscript to reflect our new data with pyrin/NACHT-spanning qPCR primers, and explained this qPCR strategy more fully in the figure 4A caption.

3. We now confirm that our epithelial cell lines do not express NLRP3 protein by western blot, relative to positive control samples (**new Figs 4B, 4F**).
4. We believe that the above data provide compelling evidence that epithelial cells do not express NLRP3 and so cannot signal via this inflammasome. However, to provide further evidence that LPS-induced IL-1 β responses in these cells are fully independent of NLRP3, we pre-treated cells with the NLRP3 inhibitor MCC950 that ablates NLRP3 responses (**Fig. 3A-C**). Indeed, LPS-induced IL-1 β secretion in both HBEC and HIEC

cells was insensitive to MCC950 and thus NLRP3-independent (Figs 4C-E, 4G-H). In keeping with an inability of epithelial cells to signal via NLRP3, HIEC-6 cells were unable to mount an IL-1 β or cell death response to the prototypical NLRP3 activator, nigericin (Fig 4G-H).

Issue 3: Throughout the manuscript, the authors conclusions are not directly supported by the evidence provided and some leaps in logic are implied. Revising the writing would help the reader.

As suggested by the reviewer we have substantially revised the manuscript text to increase clarity and ensure logical transitions.

Example 1, Abstract: "although the molecular mechanisms that activate caspase-4 and govern its capacity to cleave substrates are poorly defined". The authors' current and previous papers address the activation of caspase but not regulation of substrate interaction or substrate selection. Therefore, "the capacity to cleave substrates" has not been addressed.

Since a caspase zymogen can only cleave its substrate after it acquires protease activity, caspase-4 activation is intrinsically linked to its capacity to cleave substrates. As detailed above, the current manuscript focuses on how caspase-4 dimerisation and IDL self-cleavage allows this protease to cleave its substrates GSDMD and pro-IL-1 β . Thus, we feel that this text provides an appropriate introductory rationale for the study.

Example 2, Abstract: "acquires protease activity within the non-canonical inflammasome by forming a dimer that self-cleaves at D285 to directly cleave GSDMD". Auto-activation of caspase by cleavage at D285 does not inform us that subsequent cleavage of GSDMD is "direct". The evidence provided only supports that auto-cleavage of the caspase at D285 is necessary for GSDMD cleavage.

As requested, we have removed the word "directly" from this sentence, which referred to earlier work from us and others demonstrating that caspase-11 cleaves GSDMD.

Reviewer #3:

In this manuscript, Chan et al, by using the DmrB dimerization system, found that caspase-4 dimerizes then self-cleaves at two sites - D270 and D289 - in the interdomain linker(IDL) to drive GSDMD-dependent cell lysis. Interestingly, the author also reports that self-cleavage at D289 generates a caspase-4 p34 form that directly cleaves pro-IL-1b. This is in contrast to its murine orthologue, caspase-11, which doesn't directly cleave pro-IL-1 β into its mature form, but relies on the associated NLRP3-activation-to drive caspase-1 dependent cleave of pro-IL-1b. Using HMDMs and HBEC-KT cells, the author further found that Cytosolic LPS induces GSDMD and IL-1 β processing independently of NLRP3 or caspase-1. Overall, this is a good-quality study; the experiments are well performed and the data are strong, providing a new perspective to understanding the mechanism of activation of caspase-4. However, there are still a few gaps and limitations that need to be addressed in order to firmly establish the mechanism proposed by the authors.

We thank the Reviewer for their positive comments and support for our study. The revised manuscript provides substantial further evidence supporting our proposed mechanism of

caspase-4 activation and direct pro-IL-1 β cleavage, as detailed in the points below.

The authors set up a robust in vitro assay to examine the activation of CASP4. We can clearly see that the Cytosolic LPS induces GSDMD-pyroptosis and pro-IL-1 β cleavage independently of the NLRP3 inflammasome. However, the current data have not firmly established direct evidence that Caspase-4 dimerizes and directly cleaves pro-IL-1 β is lacking:
- In fig 3 can the author examine the oligomerization/ dimerization of caspase-4 using the native gel?

Our revised manuscript includes seven new data panels, which alongside our original data, provides a compelling case for direct cleavage of pro-IL-1 β by caspase-4. We attempted to examine caspase-4 dimerisation/multimerization through biochemical approaches (crosslinking, native PAGE), but these experiments yielded data that were difficult to interpret – likely because caspase-4 interacts with a multitude of accessory proteins (e.g. GBPs) during LPS sensing, and upon dimerisation self-cleaves at two residues – and such approaches did not allow us to rigorously identify dimeric species on the basis of size.

We have thus used an extensive suite of alternative methods to demonstrate that caspase-4 directly generates the mature, secreted form of IL-1 β independently of the NLRP3 inflammasome:

1. We engineered a DmrB-caspase-4 fusion protein to precisely control caspase-4 dimerisation, auto-processing and substrate cleavage in cells; using this system we showed that:
 - a. Dimeric caspase-4 auto-processed at the D289 residue (p34/p9 species) induces the maturation of human pro-IL-1 β (**Fig 2B**)
 - b. Caspase-4 dimers require the catalytic cysteine of their protease domain (C285) in order to induce cleavage of human and murine pro-IL-1 β (**new Fig 2C**), confirming that the protease activity of caspase-4 drives pro-IL-1 β cleavage.
 - c. In side-by-side comparisons, dimeric caspase-4 and caspase-1 induce cleavage of human and murine pro-IL-1 β , while caspase-11 cannot; confirming the known capacity of caspase-1 but not caspase-11 to cleave pro-IL-1 β (**new Fig 2C**), and revealing that caspase-4 has a distinct function compared to its murine ortholog caspase-11.
2. We used a fully recombinant, cell-free system to confirm that caspase-4 directly cleaves pro-IL-1 β . Here, we incubated recombinant purified caspase-4 with recombinant purified pro-IL-1 β (all sourced commercially) in a tube with nothing other than buffer. In this fully recombinant system caspase-4, like caspase-1, directly cleaved pro-IL-1 β . Given that this cleavage assay is a fully reductionist system – the only proteins present are caspase-4 and its substrate pro-IL-1 β – we consider this the gold standard assay for demonstrating that caspase-4 directly cleaves this substrate.

Can the author also use shRNA or siRNA to support the claim that caspase-4 activation in macrophages directly cleaves IL-1 β ?

We thank the Reviewer for this excellent suggestion, which we have now adopted to provide further evidence that caspase-4 directly cleaves pro-IL-1 β independently of NLRP3 in cells. Figure 3 now shows that caspase-4 is critically required for IL-1 β secretion (**Fig 3A, Fig 3C, new Fig 3F**) and cleavage (**Fig 3C**) in human monocyte-derived macrophages (HMDM) challenged with cytosolic LPS. HMDMs express NLRP3, and non-canonical inflammasome

signalling triggers a second wave of signalling by the NLRP3-ASC-caspase-1 inflammasome in these cells. Thus, to examine mechanisms of NLRP3-independent IL-1 β responses in HMDM, we treated these cells with MCC950 prior cytosolic LPS challenge to disable NLRP3-induced caspase-1 activation (verified in **Figs 3C, 3D**). We demonstrate the critical requirement for caspase-4 in this NLRP3-independent IL-1 β response using a pharmacological approach (the caspase-1/4 inhibitor VX765, which will only block caspase-4 signalling in MCC950-treated cells that have already silenced the NLRP3-ASC-caspase-1 inflammasome; **Fig 3A, Fig 3C, new Fig 3D, new Fig 3F**) as well as the suggested RNA knockdown approach (**new Figs 3E-3F**). All approaches consistently show that cytosolic LPS triggers substantial NLRP3-independent IL-1 β release, and that caspase-4 is required for this response.

- In Fig. 3C active Caspase 4 blot is needed to show the cleavage of caspase-4 in macrophages.

We thank the Reviewer for this excellent suggestion, which we have now adopted. Our manuscript revision now incorporates a **new Fig 3D**, which shows that cytosolic LPS induces caspase-4 and caspase-1 cleavage in HMDM. Cell pretreatment with MCC950 blocks caspase-1 cleavage without affecting caspase-4. As expected, caspase-4 cleavage is prevented by treating LPS+MCC950-stimulated cells with VX765.

- In Fig.4D it would be very informative if the author would include caspase1 and IL-1b in the western blot.

Fig 4D of our original submission (now **Fig 4E**) was western blot analysis of HBEC-KT cells. These cells produce IL-1 β , but only at low levels (~100pg/ml by ELISA; **Fig 4C**). We attempted to detect caspase-1 and IL-1 β in HBEC-KT by western, but these were below the detection limit of our western blot analysis methods, likely due to low-level expression in HBEC relative to macrophages (which e.g. produce ~10ng/ml IL-1 β by ELISA; **Fig 3A**).

- In Fig.1C. the recombinant caspase-4 cleaves IL-1b with slow kinetic? Why is that? Also, is it possible to examine this using 293T cell overexpression of full-Caspase 4 and IL-1b

We believe the Reviewer is referring to Fig 2C of our original submission (now **Fig 2D**). We do not know why caspase-4 cleaves pro-IL-1 β with a slower kinetic than human caspase-1 in a fully recombinant system, but we note similar published findings for caspase-4 cleavage of murine pro-IL-1 β (Devant 2021). We do not see the same striking difference between caspase-4 and -1 in our cellular systems (**Fig 2B, new Fig 2C**). It is possible that key regulatory mechanisms that control caspase-4 ICE activity in cells are absent from this recombinant system (e.g. caspase-4 or pro-IL-1 β post-translational modifications, substrate trafficking mechanisms), leading to less efficient ICE activity in a recombinant setup.

We thank the Reviewer for the excellent suggestion to examine caspase-1- versus caspase-4-driven pro-IL-1 β cleavage in overexpressing HEK293T, which we have adopted and extended to also include caspase-11 (**new Fig 2C**). In side-by-side comparisons, dimeric caspase-4 and caspase-1 induce cleavage of human and murine pro-IL-1 β , while caspase-11 cannot; confirming the known capacity of caspase-1 but not caspase-11 to cleave pro-IL-1 β and revealing caspase-4 as a new IL-1 β -converting enzyme (**new Fig 2C**).

A previous study (Ref Wang et al., 2020) has already shown that D289 cleavage is necessary

for LPS-induced caspase-4 signaling and GSDMD processing, whereas D270 is partially dispensable. The authors mentioned this in the discussion but should credit more the previous study and explain it in more detail in the introduction section.

As suggested, our revised manuscript now more fully discusses the Wang 2020 study in the introduction (lines 66-69) and discussion (lines 263-271). Importantly, this published work does not investigate several aspects of caspase-4 activation and function that are revealed in our current manuscript; for example: (1) the requirement for caspase-4 to dimerise to gain proteolytic activity, (2) the ability for caspase-4 to directly cleave pro-IL-1 β and the auto-processing sites within caspase-4 required for this activity; and (3) the cellular contexts in which NLRP3-independent IL-1 β maturation by caspase-4 may contribute to physiological host responses.

July 24, 2023

RE: Life Science Alliance Manuscript #LSA-2023-01908R

Prof. Kate Schroder
University of Queensland
Institute for Molecular Bioscience
St Lucia
The University of QLD
Brisbane 4072
Australia

Dear Dr. Schroder,

Thank you for submitting your revised manuscript entitled "Caspase-4 dimerisation and D289 auto-processing elicits an interleukin-1 β converting enzyme". We would be happy to publish your paper in Life Science Alliance pending final revisions necessary to meet our formatting guidelines.

- please notice Reviewer 3's minor remaining comment
- please add your main and supplementary figure legends to the main manuscript text after the references section
- please add a conflict of interest statement to your main manuscript text
- please add a callout for Figure 2C to your main manuscript text

A. FINAL FILES:

B. MANUSCRIPT ORGANIZATION AND FORMATTING:

**Submission of a paper that does not conform to Life Science Alliance guidelines will delay the acceptance of your

manuscript.**

The license to publish form must be signed before your manuscript can be sent to production. A link to the electronic license to publish form will be sent to the corresponding author only. Please take a moment to check your funder requirements.

Sincerely,

Reviewer #2 (Comments to the Authors (Required)):

I would like to congratulate the authors on providing a compelling, thoughtful and thorough revision of their work. All of my previous comments are satisfactorily addressed in the revised manuscript and the manuscript should be published with no further delay.

Reviewer #3 (Comments to the Authors (Required)):

I commend the authors on their outstanding study. All of my concerns have been addressed satisfactorily. However, I would like to point out a minor typo in the legend of Figure 1. "C258A" is mistakenly written as "C285A."

July 27, 2023

RE: Life Science Alliance Manuscript #LSA-2023-01908RR

Prof. Kate Schroder
University of Queensland
Institute for Molecular Bioscience
St Lucia
The University of QLD
Brisbane 4072
Australia

Dear Dr. Schroder,

Thank you for submitting your Research Article entitled "Caspase-4 dimerisation and D289 auto-processing elicits an interleukin-1 β converting enzyme". It is a pleasure to let you know that your manuscript is now accepted for publication in Life Science Alliance. Congratulations on this interesting work.

DISTRIBUTION OF MATERIALS:

Again, congratulations on a very nice paper. I hope you found the review process to be constructive and are pleased with how the manuscript was handled editorially. We look forward to future exciting submissions from your lab.

Sincerely,
